# Cross-sectional experimental assessment of pain modulation as part of multidimensional profiling of people with cervicogenic headache: protocol for a feasibility study

Sarah Mingels [1,2] Marita Granitzer,[2] Annina Schmid,[3] Thomas Graven-Nielsen,[4] Wim Dankaerts[1]

For numbered affiliations see end of article.

**Correspondence to**
Dr Sarah Mingels;
sarah.mingels@kuleuven.be

## ABSTRACT

**Background** An endogenous pain modulation profile, reflecting antinociceptive and pronociceptive mechanisms, may help to direct management by targeting the involved pain mechanism. For individuals with cervicogenic headache (CeH), the characteristics of such profiles were never investigated. However, the individual nature of experiencing pain demands profiling within a multidimensional framework including psychosocial lifestyle characteristics. The objective of the current protocol is to assess the pain modulation profile, which includes psychosocial lifestyle characteristics among people with CeH.

**Methods and analysis** A protocol is described to map pain modulation profiles in people with CeH. A cross-sectional non-randomised experimental design will be used to assess feasibility of mapping these profiles. The pain modulation profile is composed based on results on the Depression, Anxiety, Stress Scale, Pittsburgh Sleep Quality Index, Headache Impact Test and on responses to temporal summation of pain (pinprick), conditioned pain modulation and widespread hyperalgesia (mechanical pressure pain threshold and cuff algometry). Primary analyses will report results relating to outcomes on feasibility. Secondary analyses will involve an analysis of proportions (%) of the different psychosocial lifestyle profiles and pain profiles.

**Ethics and dissemination** Ethical approval was granted by the Ethics Committee Research UZ/KU Leuven (Registration number B3222024001434) on 30 May 2024. Results will be published in peer-reviewed journals, at scientific conferences and, through press releases. Protocol V.3. protocol date: 3 June 2024.

## STRENGTHS AND LIMITATIONS OF THIS STUDY

⇒ An innovative protocol is developed to assess facilitated pain processing in people with cervicogenic headache.
⇒ Multiple dimensions (pain processing, psychosocial lifestyle) of cervicogenic headache will be explored.
⇒ No biomarkers exist to determine facilitated pain processing. The pain status can only be estimated through proxy measures such as the conditioned pain modulation and widespread hyperalgesia.
⇒ If successful, the protocol can be adapted to increase clinical applicability.

## INTRODUCTION

Cervicogenic headache (CeH) is a secondary headache attributed to dysfunctions of the cervical spine.[1] Underdiagnosed and undertreated, CeH might evolve into a chronic state, increasing the odds of absenteeism and disability.[2 3] It is generally accepted to adopt a multidimensional approach in the management of pain.[4 5] Interestingly though, such an approach is not recommended when managing people with CeH.[6] Results from a Delphi-study indicated that lifestyle advice, pain education and cognitive therapy were considered not to be relevant in the management of CeH.[6] Currently, the non-pharmacological management of CeH mainly focusses on targeting the musculoskeletal dysfunctions of the (upper) cervical spine.[6–10] However, a meta-analysis with pooled outcome parameters showed inconsistent results of such management on headache intensity, frequency and related disability.[11] Although non-pharmacological interventions may play an important role in managing CeH, there is currently limited scientific evidence to fully support these interventions.[11] It has been stated previously that some therapeutic interventions are not appropriate for all people with CeH.[12]

When CeH is merely mediated by a peripheral nociceptive source (ie, musculoskeletal dysfunction of the upper cervical spine), also known as bottom-up source, this could be managed by addressing the dysfunction

through manual therapy and/or specified exercises.[13] Such management is likely to be inefficient if signs of facilitated central pain mechanisms are present. In such state, exclusively aiming at the peripheral source might act as a nociceptive stimulus maintaining facilitated central pain processing[13 14] The pathophysiology of CeH can thus generally be explained by (a) CeH caused by an exclusive peripheral input or (b) CeH caused by peripheral input and maintained by sensitisation processes. It has been argued that characterising the involved dominant central pain mechanism might provide valuable information to increase therapy efficacy.[15] Central pain mechanisms such as pronociceptive or antinociceptive mechanisms[16] have, however, not extensively been examined in people with CeH. Results from a study on pain processing, showing lower extracephalic and cephalic pressure pain thresholds (PPTs) in people with CeH compared with healthy controls, might indicate a dysfunctional central pain mechanism.[17 18]

Dynamic protocols, to evoke inhibition or facilitation of pain, have been designed to evaluate endogenous pain modulation and to define a pronociceptive or antinociceptive central pain mechanism. Protocols to assess pain inhibition evaluate the conditioned pain modulation (CPM) paradigm, and pain facilitation assesses temporal summation of pain (TSP).[16] The term CPM was created for psychophysical protocols that explore diffuse noxious inhibitory control in humans.[19] The latter manifests as the inhibition of wide dynamic range neuronal activity by descending adrenergic pathways.[19] CPM is measured by comparing pain induced by a test stimulus, with pain induced by the same test stimulus either during (parallel) or after (sequential) a conditioning stimulus. TSP is a nociceptive mediated process, which is considered to reflect advanced spinal synaptic facilitation in the dorsal horn (ie, behavioural correlate of wind-up).[20 21] TSP is measured by comparing pain ratings between a single noxious stimulus and repeated equal-intensity noxious stimuli at a specific frequency. A third proxy to examine facilitated central pain mechanisms is widespread hyperalgesia.[22]

Assessing endogenous pain modulation and phenotyping[23–25] in people with CeH within a multidimensional framework is needed in the context of therapy unresponsiveness. Despite the well-known pathophysiology of CeH, the number of non-responders to non-pharmacological non-invasive therapy amounts to 25%, and self-reported effectiveness of manual therapy is rated as 36%.[26 27] Such therapy unresponsiveness has in other musculoskeletal disorders been related to inadequate health literacy, neural sensitivity or augmented pain processing in the central nervous system.[26 27] Therefore, a pain modulation profile (PMP) needs to be composed. The PMP includes, besides measurements to analyse central pain mechanisms, also measurements of potential influential factors (eg, demographic, psycho social lifestyle) that can influence such mechanisms. These factors might explain some of the interindividual variability in pain perception, and,

therefore, possibly also play a role in CPM.[28] Nociceptive inputs activate complex interactions among cortical regions that are also active in cognitive, emotional and reward functions. These regions influence serotonergic and noradrenergic descending pain modulatory systems bimodally via processes between the periaqueductal grey, rostral ventromedial medulla and pontine noradrenergic nuclei, ultimately facilitating or inhibiting nociceptive input. Descending pain modulatory pathways can, therefore, be stimulated from the top-down, that is, from the brain to brainstem, by psychosocial interventions.[29–31] Furthermore, also lifestyle factors can influence pain processing. Preliminary evidence is provided that an active lifestyle could reduce spinal nociception (ie, nociceptive flexion reflex) in healthy adults.[30] Animal studies indicated that regular physical activity influences central cellular processes (ie, decrease neuronal excitability, alter neuroimmune signalling, increase release of endogenous opioids and serotonin in the descending pain modulatory pathways) involved in dysregulation of endogenous pain modulatory system and development of chronic pain.[32–34] Additionally, one night of total sleep deprivation impaired descending pain modulatory pathways, facilitated spinal neuronal excitability and facilitated peripheral pain mechanisms in heathy participants.[35]

The objective of the current paper is to describe a study protocol that aims to feasibly assess the PMP as part of multidimensional profiling among people with CeH. The advantage of determining such PMP is that it offers the possibility to address the involved pain mechanism(s) within a multidimensional framework.

## METHODS AND ANALYSIS
### Design
A first step towards achieving the objective was to describe a protocol to feasibly assess PMPs among participants with CeH. The next step will be to conduct a feasibility study. The protocol (ie, eligibility criteria, set-up, analyses of outcomes) will be adapted based on the findings and feedback gathered from the feasibility study. Relevant adaptations will be communicated with the principal investigator (professor W Dankaerts) and the Ethics Committee Research UZ / KU Leuven.

The measurements and instruments were selected based on a literature review, and after consensus with an expert panel (n=5) consisting of experts in neurophysiology (MG), neuroscience (AS), biomedical science, engineering and medical science (TG-N) and musculoskeletal rehabilitation (WD, SM) (October 2022–February 2023).

A description of the measurements, outcomes, instruments and procedure is outlined below (table 1 and paragraph 2.5). Enrolled participants will complete the protocol and will be profiled based on their PMP. This profile comprises a pain profile and psychosocial lifestyle profile. A cross-sectional non-randomised design will be used to determine if PMPs can be feasibility assessed.

**Table 1** Summary of the measurements, outcomes and instruments to feasibly assess PMPs

| Primary outcome | Instrument | Measurement |
|---|---|---|
| Feasibility | Personalised questionnaire UEQ[88] | Process metrics<br>Resource metrics<br>Management metrics<br>Scientific metrics<br>Operational feasibility |
| Profiling-related outcome | Instrument | Measurement |
| Sociodemographics | Questionnaire | Age–gender–body mass index<br>Socio-economic status |
| Headache characteristics | Headache-diary | Intensity–frequency–duration–medication intake |
| Pain profile | Pressure cuff<br>Algometer<br>Pinprick stimulator | Conditioned pain modulation (conditioning stimulus)<br>Conditioned pain modulation (test stimulus)<br>Widespread hyperalgesia<br>Temporal summation |
| Psycho-social lifestyle profile | DASS-21<br>PSQI<br>HIT-6 | Depression, anxiety, stress<br>Sleep quality<br>Quality of life |

DASS-21, Depression, Anxiety, Stress Scale; HIT-6, Headache Impact Test; PMP, pain modulation profile.; PSQI, Pittsburgh Sleep Quality Index; UEQ, User Experience Questionnaire.

### Study population and setting

Participants with CeH will be screened and recruited via the neurological staff of the headache department of the AZ Vesalius hospitals (Tongeren and Bilzen, Belgium).

A detailed summary of the study population's inclusion and exclusion criteria is provided in online supplemental table S1. A 4-week headache-diary will be distributed among the enrolled participants to collect information on the following headache characteristics: intensity, frequency, duration and medication intake. Eligible participants will need to read and sign the informed consent (collected by the principal researcher) before officially being enrolled. The experimental set-up will be at the Leuven University (Leuven, Belgium) and private practice of the principal researcher (Riemst, Belgium).

### Patient and public involvement

People with CeH will be involved in the feasibility study. This study aims to gather information on arguments to participate or decline to participate, reason(s) for premature ending, appropriateness of the measurements, and barriers, adverse events or burden experienced by people with CeH. The information will be used to adapt the protocol. Feedback on possible adaptations will be provided.

### Experimental protocol

A PMP, consisting of a psychosocial lifestyle profile and pain profile, will be composed (figure 1). Three questionnaires will be used to estimate the psychosocial lifestyle profile. The pain profile will be interpreted based on three parameters (TSP, CPM, widespread hyperalgesia).

Assessment will be in a headache-free phase (=score of 0 on the Numeric Pain Rating Scale). Participants will be informed on possible adverse events and asked to abstain from vigorous physical activity, taking analgesics and caffeine-containing beverages 24 hours prior to the measurements. Prophylactic treatment(s) remain unchanged.[1] Adverse events will be questioned after finalising the test procedures and again after 24 hours. Measurements will be performed by the principal researcher (manual therapist, >10 years clinical experience, PhD physiotherapy and rehabilitation sciences) in a chronological order (figures 2–4). The entire procedure is estimated to amount to 4 hours. An overview of the time schedule for the feasibility study is provide in online supplemental figure S1.

### Psychosocial lifestyle profile—questionnaires

The Pittsburgh Sleep Quality Index (PSQI), Depression, Anxiety, Stress Scale (DASS-21) and Headache Impact Test (HIT-6) will be completed in a quiet room prior to the CPM, TSP and PPT measurements. Thereafter the User Experience Questionnaire will be filled in. The principal researcher will be available in case of ambiguity about the questions/statements.

### Pain profile—TSP

Figure 2 provides a summary of the TSP protocol.

### Pain profile—CPM

Figure 3 provides a summary of the CPM protocol that will be executed 15 min after the TSP measurements.

### Pain profile—widespread hyperalgesia

Figure 4 provides a summary of the protocol to evaluate widespread hyperalgesia, which will be executed 15 min after the CPM measurements (online supplemental figure S2).

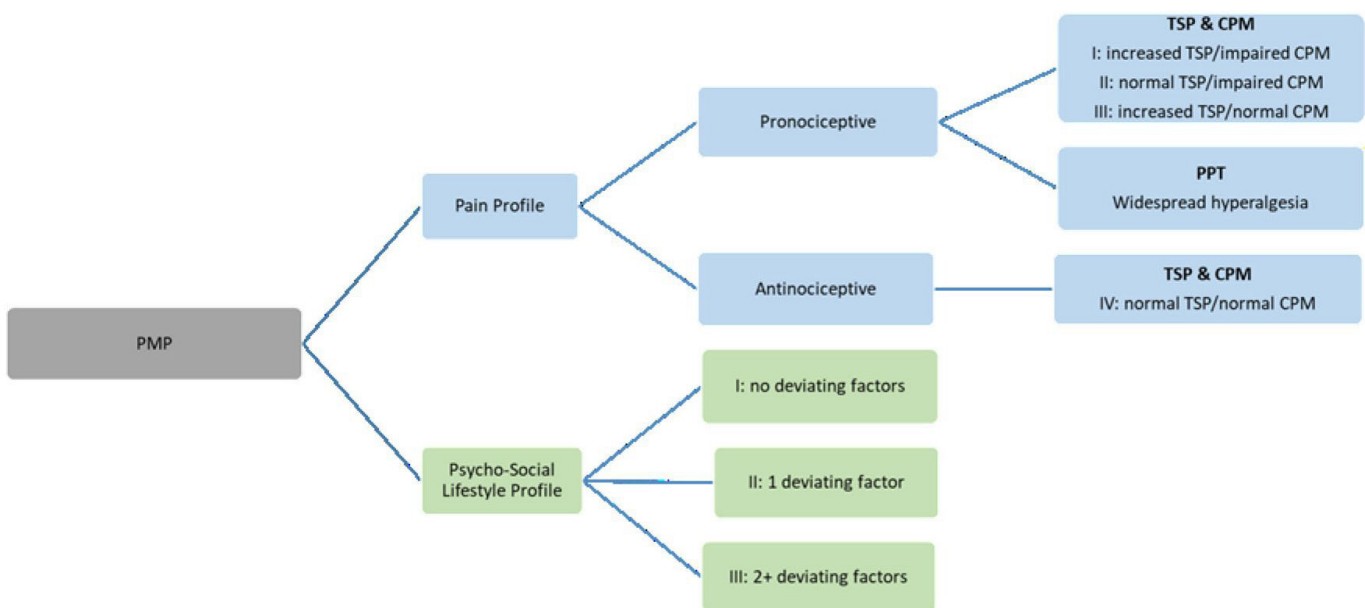

**Figure 1** Visualisation of the composition of the PMP. CPM, conditioned pain modulation; PMP, pain modulation profile; PPT, pressure pain threshold; TSP, temporal summation of pain.

## Measurements, outcomes, instruments and procedure
### Primary outcome—feasibility
Feasibility of the protocol will be assessed using adapted feasibility metrics. These metrics include process, resource, management and scientific metrics.[36 37] Details on the primary outcomes are summarised in online supplemental table S2.

### Profiling-related outcome—psychosocial lifestyle profile
Participants will be categorised in psychosocial lifestyle profile based on the number of deviating psychosocial lifestyle factors deducted from scores on the PSQI, DASS-21 and HIT-6. Individual scores will be compared with normative data.[17] Scores indicating at least: moderate depression, anxiety, stress (DASS-21), headache has a significant impact on daily life (HIT-6), and/or poor sleep quality (PSQI) will be considered as deviating.

### Psychosocial lifestyle factors
*Sleep quality* will be assessed via the Dutch PSQI, a standardised, valid, and reliable self-reported 1 month recall questionnaire.[38 39] The index differentiates poor from good sleepers by measuring: subjective sleep quality, sleep latency, sleep duration, habitual sleep efficiency, sleep disturbances, use of sleeping medication, and daytime dysfunction. Scores on each of these components vary from 0 ('No problem') to 3 ('Serious problem'). A maximum score exceeding 5/21 indicates poor sleep quality.[40 41] See Mollayeva *et al* for information on the psychometric properties.[38]

The degree of *depression, anxiety and stress* will be estimated by the Dutch DASS-21.[42 43] The DASS-21 is a self-reported, 1-week recall questionnaire. Each of the three subscales contains seven items. The depression subscales assess dysphoria, hopelessness, devaluation of life, self-deprecation, lack of interest, anhedonia and inertia. The anxiety subscale estimates autonomic arousal, skeletal muscle effects, situational anxiety and subjective experience of anxious affect. The stress subscale evaluates difficulty in relaxing, nervous arousal, being easily upset and impatience. Items are scored on a Likert scale (0='did

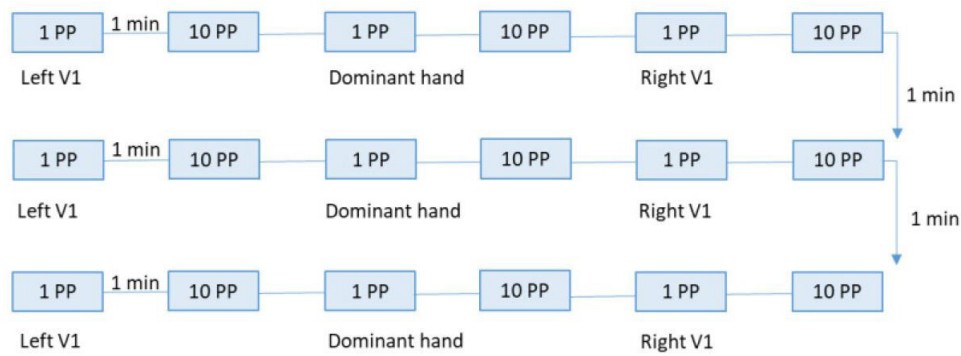

**Figure 2** Visualisation of the experimental protocol regarding TSP. PP, PinPrick; TSP, temporal summation of pain; V1, ophthalmic branch trigeminal nerve.

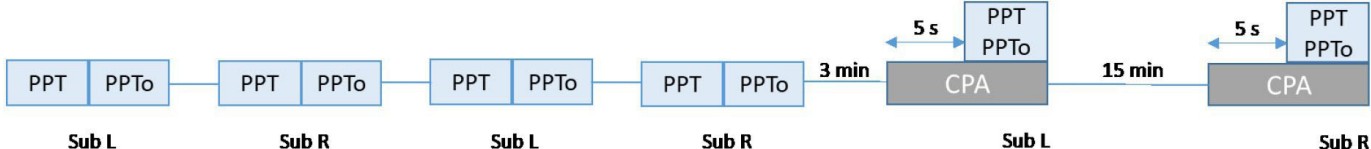

**Figure 3** Visualisation of the experimental protocol regarding CPM. CPA, cuff pressure algometry non-dominant calf muscle (kPa); L, left; PPT, pressure pain threshold (kPa/cm²); PPTo, pressure pain tolerance (kPa/cm²); R, right; Sub, suboccipital.

not apply to me at all' and 3='applied to me very much or most of the time'). Scores of 14, 10 and 19 indicate at least moderate depression, anxiety and stress, respectively. See Lovibond *et al* for information on the psychometric properties.[44]

Impact *of headache on quality of life* will be estimated by the Dutch HIT-6.[45 46] The HIT-6 evaluates the impact of headache on daily activities: the ability to function at work, school, home and in social situations. Items are scored 6, 8, 10, 11 and 13 (6='never', 8='rarely', 10='sometimes', 11='very often' and 13='always'). Scores exceeding 56 indicate headache has a significant impact on daily life.[46 47] See Martin *et al* and Kosinski *et al* for information on the psychometric properties.[46 48]

### Profiling-related outcome—pain profile
Participants will be categorised in four pain profiles based on responses to the TSP and CPM (figures 1–3):

P-P I: pronociceptive (increased TSP/impaired CPM).
P-P II: pronociceptive (normal TSP/impaired CPM).
P-P III: pronociceptive (increased TSP/normal CPM).
P-P IV: antinociceptive (normal TSP/normal CPM).

Each participant with CeH will be categorised into a pain profile per measurement location (TSP at the hand and face, CPM at the left and right suboccipital muscles).[49 50] Additional presence of widespread pain will be evaluated (figure 4). The paragraphs below contain information on the definitions of increased/decreased TSP, impaired/normal CPM and presence of widespread hyperalgesia.

#### Pinprick stimulus: TSP
The pinprick evoked Wind-Up Ratio (WUR) (Pinprick stimulator, MRC Systems GmbH—Medizintechnische Systeme) is suggested to evaluate TSP. Participants will be positioned in sitting. The WUR will be assessed at the bilateral (ie, painful and pain-free sides) ophthalmic region (V1), and extrasegmentally on the thenar eminence of the dominant hand.[49 51–53] WUR represents the quotient of pain intensity evoked by one single pinprick stimulus divided by the average pain evoked by 10 repetitive pinprick stimuli.[51–53] We additionally propose to apply the SumSquare method as described by Allison *et al* to quantifying TSP.[54] The WUR is reported to lack sensitivity to detect differences in the magnitude of sensitisation. Most studies report little variability (ratio 1.6–1.7) in WUR measurements both in normal and hyperalgesic states.[54]

For the measurements at the face and hand, constant pinpricks of 128 mN and 256 mN, respectively will be used.[52] First, a single pinprick stimulus will be applied, followed by a series of 10 pinprick stimuli with a frequency of 1 Hz within an area of 1 cm².[50 52] The interval between the single and repetitive stimuli will be 1 min, or until all aftersensations have resolved. Participants will be asked to rate the pain of a single pinprick stimulus, and the average pain at the end of the series of 10 pinpricks using the 0–10 Numeric Pain Rating Scale (0=no pain, 10=the worst pain imaginable). The test order will be standardised as: dominant hand—left side face—right side face. This procedure will be repeated three times at each site, and three measurements per site will be averaged (figure 2). We refer to the normative values provided by the German Research Network on Neuropathic Pain to determine normal or increased TSP at the face and hand.[51] Test–retest reliability for pinprick induced TSP is moderate-to-good in healthy participants (ICC 0.51–0.61).[55]

#### Test stimulus: mechanical PPT and tolerance
Measurements will be performed 15 min after the TSP measurements. PPT and pressure pain tolerance (PPTo) (kPa/cm²) will be measured at the left and right suboccipital muscles using an electronic pressure algometer (Somedic AB, Stockholm, Sweden).[56] Pressure will be perpendicularly applied directly on the suboccipital muscles, starting at 0 to maximal 1000 kPa, using a 1 cm² probe with a slope of 30 kPa/s.[17 57] Participants will be instructed to push the stop button when the sensation of pressure first elicits pain (PPT), and when the maximum amount of pressure that could be tolerated is reached (PPTo). A trial will be performed once on the left shin (tibialis anterior muscle belly, approximately 2.5 cm lateral and 5 cm inferior to the tibial tubercle) before measuring the suboccipital muscles.[58] Intrarater reliability of tibialis anterior PPT-measurements is good-to-excellent in people with CeH (ICC 0.82–0.92).[59]

Two repeated measurements at baseline, and two measurements parallel with the conditioning stimulus will be performed (figure 3). Averages of the extracted parameters will be used for further analysis.[60] Test–retest

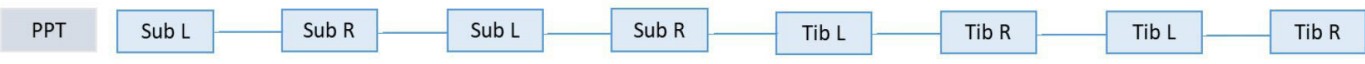

**Figure 4** Visualisation of the experimental protocol regarding widespread pain. PPT, pressure pain threshold (kPa/cm²); L, left; R, right; Sub, suboccipital; Tib, tibialis anterior.

reliability of PPT-measurements at the suboccipital muscles is good to excellent (ICC 0.84–0.93).[61] Intra-rater reliability of such PPT measurements is moderate-to-good in people with CeH (ICC 0.69–0.87).[17 62]

### Conditioning stimulus: Cuff pressure algometry

Cuff pressure tolerance (kPa) will be measured at the non-dominant lower leg. A cuff (Nocitech, Aalborg University, Denmark) will be placed around the bare non-dominant calf muscle, mounted with 8 cm distance between its upper border and the tibial tuberosity.[63 64] Upper and lower borders of the cuff will be marked on the skin to ensure the cuff does not move between stimulations. Pressure tolerance will be determined through the maximal amount of pressure that can be tolerated. Intensity of the cuff pressure for the conditioning stimulus will be predefined as 70% of the pressure tolerance on the non-dominant leg.[63–65] Computer-controlled cuff algometry (LabBench CPAR+instrument) shows an excellent intrarater reliability (ICC 0.89) in healthy subjects.[66 67] Test–retest reliability is good (ICCs 0.74–0.87) in healthy subjects.[63]

### Conditioned pain modulation

CPM will be assessed 3 min after determining cuff pressure tolerance. The pressure (conditioning stimulus) will be kept constant throughout the CPM protocol.[64 65] Five seconds after inflation of the cuff, PPT and PPTo will be reassessed as described above. Participants will be informed that the conditioning stimulus will be moderately painful, but that they should focus their attention on the test stimulus. The procedure described above will be executed two times sequentially, that is, once with the PPT measurement at the left suboccipital muscles, and once with the PPT measurement at the right suboccipital muscles as test stimulus. A 15 min interval will be provided between the CPM measurements since CPM has a short lasting (<15 min) hypoalgesic effect (figure 3).[68–71]

At the moment, there is no consensus on a normal CPM effect. We propose a method described by Vaegter *et al* to categorise the CPM response as normal or impaired based on the within-subject coefficient of variation (=within-subject SD/within-subject mean) in PPT. Participants will be profiled as having an impaired CPM if the CPM response is less than or equal to the normal within-subject coefficient of variation in PPT between two repeated assessments, and as having normal CPM if the CPM response is greater than the normal variation plus the upper limit of the 95% CI.[60]

### Widespread hyperalgesia

Widespread hyperalgesia indicates facilitated central pain mechanisms.[72] PPT and CPM measurements will be used to estimate widespread hyperalgesia.

*PPTs* at both symptomatic (bilateral suboccipital muscles) and distant pain-free (bilateral tibialis anterior muscles) areas will be used to determine widespread hyperalgesia. PPTs lower than the 95% CI lower border bound of the normative PPTs will be considered decreased (=deviating).[17] PPTs at the suboccipitals will not be repeated since data are already collected within the context of CPM (see the Conditioned pain modulation section). PPTs of the bilateral tibialis anterior muscle will be determined as described above (see the Test stimulus: mechanical PPTs and tolerance section). The test order will be standardised as: left tibialis anterior—right tibialis anterior muscle. This procedure will be repeated two times, and measurements per site will be averaged. Averages of the extracted parameters will be used for further analysis.[60] An extrasegmental *CPM measurement* will be performed as second proxy to estimate widespread hyperalgesia (online supplemental figure S2).

### Data management plan

Data concerning feasibility, CPM, TSP and widespread hyperalgesia will be electronically collected (.xlsx). Data gathered from the paper questionnaires (PSQI, DASS-21, HIT-6) will be transferred to excel files (.xlsx). Data will be protected by pseudonymising them, restricting access and access only via the multifactor-authentication (KU Leuven Authenticator app). The principal researcher will complete the pseudonymising process and GDPR (General Data Protection Regulation) questionnaire according to the KU Leuven policy. Storing is foreseen in a folder structure (OneDrive, Microsoft 365 Apps for enterprise) to secure: availability of data, confidentiality and sharing data (https://icts.kuleuven.be/storagewijzer/nl). Finished data (ie, the dataset) will be stored at the KU Leuven Research Data Repository (https://www.kuleuven.be/rdm/en/rdr/rdr). This platform enables archiving (at the end of the project), uploading, describing and sharing research data in a legal and controlled manner. Exclusively the principal researcher and principal investigator will have access to all data, other team members will only obtain restricted access. The latter implies that data are anonymous for these team members.

All data will be kept for 10 years, conform the KU Leuven policy. Supportive arguments to keep personal data for 10 years include: verification of results and future research. Final versions of datasets will be preserved for the long term on internal KU Leuven data storage facilities.

### Statistics

Primary analyses will report results relating to outcomes on feasibility. Descriptive statistics and content analysis will be used to assess feasibility of the protocol. The feasibility study will provide useful information with regard to planning, testing study procedures (eg, estimation of the recruitment rate, plausibility of multicentre collaborations, etc), and investigating outcomes to support adequate sample size calculation.[73] Data from the CMP and TSP measurements will be used to estimate post hoc sample size (G*Power V.3.1.9.4, Kiel Germany) for a larger trial.

Individual PMPs of at least 12 participants with CeH will be mapped based on the methods outlined in the Profiling-related outcome—psychosocial lifestyle profile and conditioned pain modulation sections. Secondary analyses will involve an analysis of proportions (%) of the different psychosocial lifestyle profiles and pain profiles. Currently no data are available in people with CeH, therefore a pragmatic approach is proposed based on the work of Julious who recommended 12 participants as the minimum for pilot studies.[74] Justifications for this sample size are based on rationale about feasibility and precision about the mean and variance.[74] Both primary and secondary analyses will be used to decide whether the feasibility study should be expanded to a full-scale study. The latter implies registering of the trial in accordance with the WHO Trial Registration Data Set (V.1.3.1).

## ETHICS AND DISSEMINATION

All test procedures involving human participants will be in accordance with the ethical standards of the institutional research committees and with the 1964 Helsinki Declaration and its later amendments. Ethical approval was granted by the Ethics Committee Research UZ/KU Leuven (Registration number B3222024001434) on 30 May 2024. Results will be published in peer-reviewed journals, at scientific conferences, and through press releases.

## DISCUSSION
### Relevance of the study

'Maybe not all therapeutic interventions are appropriate for all people with CeH'.[12] With this statement, Fernández-de-Las-Peñas et al drew attention to an important problem in the management of CeH. Others have hypothesised that not all people with CeH can be managed successfully by only targeting the upper cervical spine.[12 13 15] Chua et al advised that the management of CeH should not exclusively focus not only on the cervical nociceptive source but also on central nervous system sensitisation spreading to the level of the trigeminal spinal nucleus.[75]

Findings of previous work of our group confirmed this by identifying secondary and tertiary hyperalgesia (ie, increased pain sensitivity in undamaged tissue away from, and contralateral of the nociceptive source). Meaning that both bilateral cephalic and extracephalic PPTs were lower in people with CeH compared with healthy asymptomatic controls.[17] However, these findings do not imply that peripheral nociceptive sources should be ignored. Such sources could initiate, maintain and modulate facilitated central pain processes. Therefore, potential nociceptive sources alongside psychological, behavioural and social components should also be assessed (ie, interplay of bottom-up and top-down mechanisms).[76]

Although CPM testing was already applied to people experiencing various types of headache such as chronic migraine, tension-type headache and post-traumatic headache,[18 77–79] it was only once administered to people with chronic CeH.[75] These people with chronic CeH experienced moderate pain (numeric pain rating scale 6.5±1.8) during CPM testing. We propose to analyse endogenous pain modulation during a headache-free phase. Having a less efficient CPM, when being headache free, suggests that on a pain-generating event, the person is at a higher risk to develop pain than people showing an efficient CPM at baseline.[29]

### The protocol

We propose a protocol to explore feasibility to assess PMPs in people with CeH. Interindividual variability in endogenous pain modulation is expected in these people due to their heterogeneous profile.[17 59 80 81] Reduced endogenous pain modulation might be expected in people with CeH presenting with modifiable lifestyle and/or psychosocial risk factors for chronification. This hypothesis is supported by PPTs at the suboccipital muscles being influenced by the level of physical activity, stress, quality of life and screen time in people with CeH.[17] Therefore, we will map the multidimensional PMP (ie, psychosocial lifestyle profile and pain profile) of each participant. Knowledge of a person's profile might support an individual management programme as opposed to the traditional one-size-fits-all approach.[16]

We opted to use a dynamic protocol to assess the pain profile of people with CeH rather than relying exclusively on questionnaires and/or static measurements. Caution is indicated when screening questionnaires (eg, central sensitisation inventory) are used to identify human assumed central sensitisation. These questionnaires are not associated with widespread pain sensitivity and show a stronger association with psychological measures than with signs of facilitated central pain mechanisms.[82]

### Feasibility

Feasibility of the protocol is anticipated since it is designed according to the latest literature in the domain of TSP and CMP measurements.[24 51 52 55 65 68 69 83 84] Collaboration with the AZ Vesalius hospitals (Tongeren and Bilzen, Belgium) is foreseen to obtain the required sample size of 12 people with CeH.[17 80 81]

A mitigation plan is composed to anticipate potential adverse events related to the testing procedure such as light headache, muscle soreness or sensitivity caused by mechanical pressure on cutaneous, muscular, nervous or arterial tissue. This plan includes general precautions such as informing participants on possible transient adverse effects, applying a strict protocol, using a wide cuff at the least uncomfortable test location (ie, calf muscles), and asking to abstain from vigorous physical activity at least 24 hours prior to the measurement. Potential adverse effects are expected to resolve within a 10 min recovery period.[85] This feasibility study will provide useful information with regard to tolerability, planning, testing study procedures (eg, estimation of the recruitment rate, plausibility of multicentre collaborations) and investigating outcomes to support

adequate sample size calculation.[73] Furthermore, such study is designed to ensure that a possible future larger intervention study is achievable, rigorous and economically justifiable to avoid waste of resources.[73] In the current context of a feasibility study, a data monitoring committee will not be involved. Adverse events are expected to be minor and transient in nature, and no long-term follow-up or interventions are applied.

Furthermore, measurements will be conducted with a respected interval (15 min after CPM) to limit carryover of the responses.[70 71]

## Limitations and suggestions

We acknowledge that assessing facilitated central pain mechanisms clinically is challenging for several reasons such as the absence of biomarkers to support its diagnosis. Although quantitative sensory testing is a promising surrogate measure to identify such facilitated mechanisms, it only evaluates evoked responses rather than spontaneous pain.[82] Results should be interpreted within that context.

More research is needed into relevant modifiable psychosocial and lifestyle factors concerning sensitisation of pain in CeH.[86] Sedentary time, anxiety, sleep quality, physical activity and stress were already reported to be associated with signs of central sensitisation in some people with CeH.[17] Additionally, interindividual differences explain a large proportion of CPM variance.[87] This finding forces researchers to look at the CeH population as more heterogeneous.[17 59] We propose a feasibility study, which includes only 12 participants with CeH, which is the minimum recommended for pilot studies.[74]

Furthermore, WUR is reported to lack sensitivity to detect differences in the magnitude of sensitisation. Most studies report little variability (ratio 1.6–1.7) in WUR measures both in normal and hyperalgesic states. Therefore, we additionally propose to apply the SumSquare method as described by Allison *et al*.[54] The potential future full-scale study should analyse endogenous pain modulation within a repeated-measures design to detect variance (ie, interindividual differences).[87] Finally, using CPM, TSP and widespread hyperalgesia to compose the PMP can be time-consuming, expensive, and equipment is often not readily available in clinical practice. Therefore, the current protocol is an essential step towards further research (eg, assessing feasibility, validity, different populations), including development of easy-to-use alternative valid measurements to be widely implemented in clinical settings.

## Author affiliations

[1]Musculoskeletal Research Unit, Department of Rehabilitation Sciences, Faculty of Kinesiology and Rehabilitation Sciences, Leuven University, Leuven, Belgium
[2]REVAL Rehabilitation Research Centre, Biomedical Research Institute, Faculty of Rehabilitation Sciences, Hasselt University, Hasselt, Belgium
[3]Nuffield Department of Clinical Neurosciences, University of Oxford, Oxford, UK
[4]Center for Neuroplasticity and Pain (CNAP), Department of Health Science and Technology, Aalborg University, Aalborg, Denmark

**Contributors** SM drafted the article. AS, TG-N, MG and WD revised the article critically for essential intellectual and methodological content. All authors approved the version to be published.

**Funding** AS was funded in whole, or in part, by the Wellcome Trust (222101/Z/20/Z). For the purpose of open access, the author has applied a CC BY public copyright license to any Author Accepted Manuscript version arising from this submission. TGN is a part of the Center for Neuroplasticity and Pain (CNAP) supported by the Danish National Research Foundation (DNRF121). SM is supported by the Leuven University (PDMt1/22/016). The sponsor has no role in composing the study design, collection, management, analysis and interpretation of data, writing of the report or the decision to submit the report for publication.

**Competing interests** None declared.

**Patient and public involvement** Patients and/or the public were not involved in the design, or conduct, or reporting, or dissemination plans of this research.

**Patient consent for publication** Not applicable.

**Provenance and peer review** Not commissioned; externally peer-reviewed.

**ORCID iD**
Sarah Mingels http://orcid.org/0000-0001-9024-2377

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
