## [Reviewer comments · BMJ Open]

ARTICLE DETAILS

TITLE (PROVISIONAL)	Cross-Sectional Experimental Assessment of Pain Modulation as part of Multidimensional Profiling of People with Cervicogenic Headache: Protocol for a Feasibility Study
AUTHORS	Mingels, Sarah; Granitzer, Marita; Schmid, Annina; Graven-Nielsen, Thomas; Dankaerts, Wim

VERSION 1 – REVIEW

REVIEWER	Patil, Deepali Ravi Nair Physiotherapy College
REVIEW RETURNED	28-Jul-2023

GENERAL COMMENTS	title should be complete PICO format. population.
---

REVIEWER	García, Israel Morales Hospital General de Zona #2
REVIEW RETURNED	05-Aug-2023

GENERAL COMMENTS	IT WOULD BE INTERESTING TO BE ABLE TO CARRY OUT A SCALE DERIVED FROM THIS STUDY SO THAT EVERYTHING IS SIMPLER AND TO BE ABLE TO USE IT AS A UNITARY SURVEY WORLDWIDE.
---

REVIEWER	Liang, Zhiqi The University of Queensland
REVIEW RETURNED	19-Oct-2023

GENERAL COMMENTS	Dear Authors, Congratulations on your detailed and well thought out protocol for an interesting and valuable study. I look forward to seeing the results and your future studies in this area.
---

VERSION 1 – AUTHOR RESPONSE

Reviewer 1 (R#1)

R#1 comment 1. Title should be complete PICO format.

We thank the reviewer for this comment (see also our response to EC comment 1). We think that the title 'Assessment of Endogenous Pain Modulation as part of Multidimensional Profiling of People with Cervicogenic Headache: A Protocol for a Feasibility Study' contains all relevant PICO information. P = People with cervicogenic headache, I = Assessment of endogenous pain modulation, C = Not applicable, and O = Feasibility, Pain Profiling as part of Multidimensional Profiling. If we would mention conditioned pain modulation, temporal summation of pain, widespread hyperalgesia, and all feasibility-related outcomes to the title it might be too verbose and incomprehensive.

We integrated the study objective, and added the setting (experimental) and design (cross-sectional). We also added 'as part of multidimensional profiling' to the objective of the study to stress that pain modulation profiling is only one part of comprehensive multidimensional profiling.

Original title: 'Assessment of Endogenous Pain Modulation as part of Multidimensional Profiling of People with Cervicogenic Headache: A Protocol for a Feasibility Study'

Adapted title: 'Cross-Sectional Experimental Assessment of Pain Modulation as part of Multidimensional Profiling of People with Cervicogenic Headache: Protocol for a Feasibility Study'

Reviewer 2 (R#2)

R#2 Comment 1. It would be interesting to be able to carry out a scale derived from this study so that everything is simpler and to be able to use it as a unitary survey worldwide.

Indeed, we agree with the reviewer. However, several steps need to be taken before clinically implementing our protocol (e.g., assessing feasibility, validity, different populations, and providing easy-to-use alternative valid measurements).

We added to:

- 'Limitations and suggestions': 'Finally, using CPM, TSP, and widespread hyperalgesia to compose the PMP can be time-consuming, expensive, and equipment is often not readily available in clinical practice. Therefore, the current protocol is an essential step towards further research (e.g., assessing feasibility, validity, different populations), including development of easy-to-use alternative valid measurements to be widely implemented in clinical settings.'

- 'Strengths and limitations of this study': 'If successful, the protocol can be adapted to increase clinical applicability'